# Tocochromanols in the Leaves of Plants in the *Hypericum* and *Clusia* Genera

**DOI:** 10.3390/molecules30030709

**Published:** 2025-02-05

**Authors:** Inga Mišina, Danija Lazdiņa, Paweł Górnaś

**Affiliations:** Institute of Horticulture, Graudu 1, LV-3701 Dobele, Latvia; inga.misina@llu.lv (I.M.); danija.lazdina@llu.lv (D.L.)

**Keywords:** John’s wort, photosynthetic, herb, lipophilic bioactive compound, vitamin E

## Abstract

Now under Clusiaceae and Hypericaceae, *Clusia* and *Hypericum* were previously categorized under one family until they were divided in 2003 by the APG III system. The *Clusia* genus is characterized by the presence of tocotrienol derivatives with antiangiogenic properties, and only *Hypericum perforatum* tocochromanol content has been studied in the *Hypericum* genus. Twelve species were analyzed: *H. aegypticum*, *H. calycinum*, *H. empetrifolium*, *H. lancasteri*, *H. olympicum* f. *minus* ‘Sulphureum’, *H. perforatum*, *H. xylosteifolium*, *C. fluminensis*, *C. minor*, *C. odorata*, *C. palmicida*, and *C. tocuchensis*. Plant leaves were analyzed for their tocochromanol (α-, β-, γ-, and δ-tocotrienol and tocopherol) contents using a reverse-phase high-performance liquid chromatography with fluorescent light detector (RP-HPLC-FLD) method. While α-tocopherol (α-T) was present in the highest proportion, the leaves had significant tocotrienol (T3) contents. Following α-T, δ-T3 was present in most *Clusia* samples, and γ-T3 in most *Hypericum* samples, except *H. olympicum*, in which α-T3 followed. *C. minor* had the highest α-T (112.72 mg 100 g^−1^) and total tocochromanol (141.43 mg 100 g^−1^) content, followed by *C. palmicida* (65.97 and 82.96 mg 100 g^−1^, respectively) and *H. olympicum* (α-T 32.08, α-T3 30.68, and total tocochromanols 89.06 mg 100 g^−1^). The *Hypericum* genus is a valuable source of tocotrienols, with potential use after purification.

## 1. Introduction

Tocochromanols are lipophilic prenyllipid antioxidants, the most common forms being tocopherols (Ts) and tocotrienols (T3s) (Figure 1). They consist of a methylated chromane ring and lipophilic prenyl lipid tail. Both are synthesized in chloroplasts and use a common biosynthetic pathway, but different chemical precursors [1]. The chromane ring of all tocochromanols is produced from homogentisate, produced in the cytoplasm. The precursor of the saturated tocopherol tail is produced from phytyl pyrophosphate, which can derive from the non-mevalonate (MEP) pathway or from degraded chlorophyll through phytol recycling. The tocotrienol tail, which has three unsaturated bonds, is produced from geranylgeranyl pyrophosphate, which is only produced in the MEP pathway and is also used in the biosynthesis of carotenoids [2] and phytyl pyrophosphate [3]. Geranylgeranyl pyrophosphate and phytyl pyrophosphate are used in the biosynthesis of chlorophylls [4]. Due to the three unsaturated bonds in the tocotrienol tail, it is significantly more lipophilic (hydrophobic), and their other chemical properties differ as well. Tocotrienols are more readily transferred between membranes, although tocopherols increase membrane rigidity more significantly [5]. In a relatively small test group, tocotrienol supplementation showed similar improvement of biological antioxidant indicators, with tocotrienols affecting female participants slightly more than male ones [6]. At high concentrations, α-tocopherol and α-tocotrienol can act as pro-oxidants. As antioxidants, their activity differs by tail and methyl group placement (homologue), while α-tocopherol is more potent at lower concentrations than γ-tocopherol, but not at high concentrations. In membranes, tocopherols and tocotrienols with the same chromanol ring structure have similar antioxidant activities, but only α-tocopherol and α-tocotrienol reduce Cu (II) to Cu (I) and showed pro-oxidant effects on methyl linoleate in sodium dodecyl sulfate micelles [5]. In food systems, tocotrienols can outperform tocopherols as antioxidants [7] and are believed to be more potent antioxidants, but direct comparisons and recent studies are scarce. 

Most notably, α-T exhibits vitamin E-vitamer activity. Ts and T3s are commonly used as antioxidants in food and cosmetics products. In terms of antioxidant activity, α-T3 appears similar or slightly better than α-T [7], although it depends significantly on the system in which it is tested and the concentration of the tocochromanol. A variety of health benefits are associated with the intake of tocochromanols, including protection against cardiovascular and neurodegenerative diseases, metabolomic and oxidative stress-related conditions, as well as anti-tumoral activity [8]. Results and conclusions are often inconclusive. Daily palm oil tocotrienol and carotene supplementation increased plasma tocotrienol and carotene levels but showed no change in cardiovascular function [9], diabetic kidney disease [10], and neuropathy [11]. Tocotrienols, δ-T3 in particular, can promote wound healing [12] by inhibiting reactive oxygen species and decreasing ferroptosis [13], a type of iron-dependent cell death.

The compounds of interest were first identified in *Hevea brasiliensis* latex, but the richest industrial and dietary sources in food include vegetable oils. The richest sources of both Ts and T3s include wheat germ oil [14], cereal bran oils, palm oil [8], berries (such as cranberries, American and European, and grapes), and seed oils [15,16]. The largest dietary tocotrienol contribution is made through cereal products [17] or products containing palm oil, depending on dietary patterns and supplement use. Official figures on the tocopherol and tocotrienol market are not available, but most supplement products are derived from palm oil, followed by annatto extracts.

Tocopherols are common in all parts of the plant and usually make up all of the tocochromanols in plant tissue, while tocotrienols are generally present in the seeds or bran of some plants [18]. If tocotrienols are present in seeds, their content decreases during germination [19]. Photosynthetic tissue tocochromanols are dominated by α-T, although fruiting bodies can contain predominantly γ-T [20]. Concentration in roots is much lower than in leaves and seeds, and α-T is also most prevalent [21,22]. Tocotrienols are rarely present (rarely recorded) in the photosynthetic parts of plants and roots, although there are reports on small amounts of α-T3 in carrot roots [23], lily flowers [24], and in various stinging nettle (*Urtica leptophylla* Kunth) plant organs [25].

*Hypericum* (St. John’s wort) and *Clusia* are genera of vascular plants belonging to the Hypericaceae and Clusiaceae families. Up until 2003, Hypericaceae was classified as a subfamily of Clusiaceae. While *Hypericum* species are present on most continents and in most climate zones, *Clusia* is only common in tropical climates. Many *Hypericum* species are used in phytomedicine for the treatment of various physical ailments [26,27] and mild to moderate depression [28], while *Clusia* is generally cultivated as a decorative plant. The compounds of primary concern in *Hypericum* species are prenylated phloroglucinols (hyperforin, adhyperforin) and polycyclic anthraquinones (hypericin, pseudohypericin). Hypericin is most often investigated as a mild antidepressant [29,30,31,32], while hyperforin is used in photodynamic therapy to treat various cancers [33,34] and as an antiviral agent [35,36,37]. Like tocotrienols, phloroglucinol and anthraquinone compounds have also been observed in *Clusia* and *Garcinia* species, including 6S,8S,28S-nemorosic acid (polyprenylated phloroglucinol) in *C. nemorosa* [38] and acylphloroglucinols in *G. paucinervis* and *G. mutiflora* [39,40]. Moreover, these genus-specific bioactive compounds share some properties with tocotrienols. Like tocotrienols, *Hypericum* extracts are considered a wound-healing-promoting agent [41]. Hypericin is considered to have protective properties against Alzheimer’s [42], and there is some evidence that tocotrienols can be helpful in treating Alzheimer’s in some patients [43], since low tocopherol and tocotrienol plasma levels have been associated with increased Alzheimer’s disease and cognitive impairment risk [44].

Multiple papers have reported the presence of tocotrienol derivatives in *Clusia* and *Garcinia* (Clusiaceae) species’ stem bark and seeds [45], as well as other former Clusiaceae members, such as *Calophyllum calaba* and *Calophyllum inophyllum* [46]. Meanwhile, phytochemical profiling in *Hypericum* has focused on hydrophilic compounds such as phenolics and flavonoids [47,48], and a study investigating lipophilic compounds in the plant only used tocopherol standards [49], while another investigating leaves noted typically α-T-dominated profiles with a small δ-T3 peak, and quantitative data were not presented [50]. Apart from these investigations of limited genus representation or limited compound representation, there has not been a dedicated study aimed at screening *Hypericum* or *Clusia* leaf tocochromanols. The first paper investigating tocochromanol content in *Hypericum* leaves was published in 2012 and observed α-T and δ-T3 in the leaves using LC-MS but did not provide exact tocochromanol contents [50]. More recently, a study investigated tocochromanol extraction from *H. perforatum* using hydroethanolic solutions and observed different results—high δ-T3 content, as well as α-T, α-T3, and γ-T [51]—that were not observed in the 2012 LC-MS study. While it did not use tocotrienol standards, another paper investigating several *Hypericum* species observed that δ-T was the main tocochromanol produced by *H. perfoliatum* (*Drosocarpium* section), *H. tomentosum* (*Adenosepalum* section), *H. ericoides* (*Coridium* section), and *H. perforatum* (*Hypericum* section) [49]. A lack of consensus on tocochromanol composition is evident in the *Hypericum* and *Clusia* genera.

The current study shows not only that *Clusia* and *Hypericum* genera are valuable sources of tocotrienols, but also shows a possible future direction for the use of *Hypericum* species, for example, in the pharmaceutical industry. Such an approach would contribute to a more efficient use of the harvested plant material St. John’s Wort and a better understanding of some of the medicinal effects it possesses.

## 2. Results and Discussion

Tocochromanol content and profile were analyzed in the leaves of seven *Hypericum* species belonging to six sections: *Adenotrias* (*H. aegyptiacum*), *Ascyreia* (*H. calycinum* and *H. lancasteri*), *Coridium* (*H. empetrifolium*), *Hypericum* (*H. perforatum*), *Inodora* (*H. xylosteifolium*), and *Olympia* (*H. olympicum* f. *minus* ‘Sulphureum’), and five *Clusia* species: *C. fluminensis*, *C. minor*, *C. odorata*, *C. palmicida*, and *C. tocuchensis*. Figure 2 shows tocochromanol RP-HPLC-FLD chromatograms in the leaves of *H. lancasteri* and in a standard solution. The chromatograms of all investigated species are presented in the Appendix A. Several compounds with typical tocochromanol UV spectra are present in the chromatograms, in addition to others that belong to other chemical classes, based on their UV spectra but which, unfortunately, could not be identified. Due to the presence of unidentified co-eluted compounds along with the four tocopherol and four tocotrienol homologs, the application of both a diode array detector (DAD) and fluorescence detector (FLD) was used to minimize misinterpretation of the chromatographic results. The tocochromanol UV spectra with the highest concentrations for each species are shown in Appendix A. It was not possible to obtain UV spectra in the case of trace and low amounts of tocochromanols. Some studies indicate that combined DAD and FLD detection can help confirm the results of FLD by the UV spectra verification, but confirmation with MS can be recommended if only trace amounts are present [52]. Considering the presence of uncommon tocochromanols in the *Clusia* genus, verification with MS is advisable in future studies.

Tocochromanol concentrations are provided in Appendix A and depicted in Figure 3. While tocochromanol contents were highly variable in both genera, α-T constituted a majority of total tocochromanols in most of the tested samples, with the highest α-T content being in *C. minor* (112.72 mg 100 g^−1^). In *Hypericum* samples, α-T was usually followed by γ-T3, highest in terms of content and ratio in *H. lancasteri* (15.92 mg 100 g^−1^ or 31% of total tocochromanols). In most *Clusia* samples, α-T was usually followed by δ-T3, highest in *C. minor* (19.52 mg 100 g^−1^) and *H. olympicum* f. *minor* ‘Sulphureum’ (17.77 mg 100 g^−1^). *H. olympicum* also had a uniquely high α-T3 content (30.68 mg 100 g^−1^) and proportion (34.47% of total tocochromanols).

Among tested *Clusia* species, *C. minor* had the highest tocochromanol content, composed mainly of α-T, followed by δ-T3. The presence of α-T in *C. minor* leaf ethyl acetate and methanol extracts was documented before with GC-MS, but δ-T3 was not identified in the study [53] and the concentration was much lower (3.29% and 2.50% of the extract, respectively).

The presence of tocotrienol derivatives in a number of *Clusia* species is comparatively well documented, including 2Z- and 2E-δ-tocotrienoloic acids in *C. criuva* [54] and *C. obdeltifolia* [55]; the former compound was also observed in *C. burlemarxii* [56]. In *Garcinia paucinervis*, another member of the Clusiaceae family, the tocotrienol derivatives paucinochymol A-F have been observed [39], as well as a tocotrienol quinone dimer in *G. nigrolineata* leaves [57]. Some δ-T3 derivatives containing garcinoic acid [58], a compound typically found in the Clusiaceae family, were also found in propolis from Colombia. In the seeds of *Calophyllum inophyllum*, a former member of the Clusiaceae family, T3 domination has been observed before, particularly δ-T3 (23.6 mg 100 g^−1^) [19]. There are no reports on non-derivative tocotrienols in the *Clusia* and *Garcinia* species, but this is not a sufficient basis for their discarding as sources of tocochromanols. The tocochromanol content, including tocotrienol, can vary considerably within the same species depending on genotype, ecotype, and variety [59]. A saponification process, which is the most effective protocol for the recovery of tocochromanols from plant material [60], was used in the current study; however, it does not discriminate between tocochromanols and their derivatives. Tocopherol and tocotrienol fatty acid esters are the most common tocochromanol derivatives reported in the existing literature and can constitute a major fraction of plant tocochromanols [61,62,63,64]. These derivatives are bonded to fatty acids by an ester bond, which is cleaved during saponification as it would be in a triglyceride. The effect of saponification on other types of derivatives is not well-studied, and tocochromanol-fatty acid esters are not the most common tocochromanol derivatives reported in Clusiaceae members. The tocotrienoloic acids present in *Clusia* species [54,55,65] may interact with the saponification process, although they are unlikely to show up in the chromatograms. While garcinoic acid, a δ-T3 derivative, is present in members of the Clusiaceae family, including some *Clusia* species, there are no reports of it in *Hypericum*, whereas the presence of some δ-T3 has been confirmed in *H. perforatum* leaves by LC-MS [50]. Although the fate of garcinoic acid and other tocotrienoloic acids during saponification was not formally demonstrated or studied, it can be assumed that, like fatty acids, it forms a salt, whereas non-derivatized tocotrienols and tocopherols are non-saponifiable.

Among *Hypericum* species, tocochromanol content was highest in *H. olympicum*, but the lack of existing reports on tocochromanols in *Hypericum* species makes the results difficult to corroborate. The second-highest TT content was observed in *H. aegyptiacum* leaves. There are no reports on tocochromanol content in most of the analyzed species. In *H. perforatum* hexane extracts, much lower α-T, δ-T, and γ-T have been observed previously, with distinct δ-T domination [49], while another study determined that *H. perforatum* leaf methanol extract contains mostly α-T and some δ-T3 [50]. A recent paper reports the occurrence of four tocopherol and four tocotrienol homologs in *H. perforatum* inflorescences, mainly δ-T3 and α-T3 [51]. Some of the differences with other studies can be explained by different extract preparation protocols—the present study extracted tocochromanols in ethanol and saponified the extract, whereas previous studies have analyzed direct extracts, which, along with a different solvent, can significantly affect the extracted tocochromanol profile. Saponification of soy ethanol extracts yielded higher tocopherol content compared to an ultrasonication-assisted protocol [66]. A similar observation was reported in grape seeds [67]. *Hypericum* inflorescences saponified in an ethanol-based environment and then extracted with a hexane–ethyl acetate mixture had similar tocochromanol recovery to direct, ultrasound-assisted extraction in 96.2% ethanol, except for α-T, α-T3, and γ-T3, which had lower recovery in the direct extraction protocol [51]. Lower extractability can be caused by the presence of bound tocochromanols, such as tocopheryl fatty acid esters in plant material [60,61,68] or non-extractable tocochromanols [60,67]. Of the observed tocochromanols, δ-T3 required the lowest concentration of ethanol in direct extraction protocols, and all tocochromanol recovery increased with higher ethanol proportions in the solvent [51]. The solubility of tocochromanols in hydroethanolic solutions is influenced by the degree of saturation and the length of the side chain of the molecule, as well as the number of methyl substituents on the chromanol ring [51,67]. Of the mentioned solvents, hexane (0.009) and ethyl acetate (0.228) have the lowest relative polarities, followed by acetone (0.355), 2-propanol (0.546), ethanol (0.654), and methanol (0.762). Efficient tocochromanol extraction (tocopherol and tocotrienol) does not appear to require the most hydrophobic solvent, but a particular polarity [66,69].

In the studied samples, α-T content ranged from 10.97 mg 100 g^−1^ in *C. fluminensis* to 112.72 mg 100 g^−1^ in *C. minor*, averaging 41.34 mg 100 g^−1^ between all samples. This is much higher than previously observed in cultivated leafy green vegetables. Typical leaf α-T content ranges from 1.39 mg 100 g^−1^ in *Heracleum moellendorffii* (Apiaceae) to 13.36 mg 100 g^−1^ in *Toona sinensis* (Meliaceae) [70], and from 0.19 mg 100 g^−1^ in *Amaranthus viridis* (green amaranth) to 18.3 mg 100 g^−1^ in *Moringa oleifera* (moringa) leaves, 1.6 mg 100 g^−1^ in *Spinacia oleracea* (spinach) leaves, and 4.6 mg 100 g^−1^ in *Brassica oleracea* (kale) leaves [71].

Although studies conducted on leaves generally investigate only tocopherols, there are reports on the presence of T3s in plant leaves. For example, Rosaceae family fruit tree leaves contain trace or small amounts of T3s; the content is typically higher at the start of vegetation, and α-T3 and δ-T3 were observed in higher concentrations than other T3s and some Ts [72]. In stinging nettle (*U. leptophylla*, Urticaceae) leaves, small amounts of β-T3 can be found as well [25]. Many of the samples had uncharacteristically high tocotrienol contents for plant leaves. The δ-T3 content in *C. minor* and *H. olympicum* is more than in palm fruit oil (0–12.3 mg 100 g^−1^) and close to coconut (20.4 mg 100 g^−1^) and *Calophyllum inophyllum* (23.6 mg 100 g^−1^) oil [19]. The tocochromanol contents observed in the present study were closer to those observed in vegetable oils—unrefined sunflower oil can contain 59–67 mg 100 g^−1^ of α-T [14,73], and wheat germ oil can contain 192 mg 100 g^−1^ of α-T [14]. The dominance of α-T is well documented in the photosynthetic parts of plants, while tocotrienols are rare and generally present in low concentrations, if detected at all [19,74]. However, leaf tocotrienol content can be relatively high as well; for example, β- and γ-T3 in *Vellozia gigantea* (Velloziaceae, monocot clade) leaves can reach approximately 7 and 6 mg 100 g^−1^, respectively [75]. The results are somewhat corroborated by the higher taxonomical categorization of the species. Both belong to the Malpighiales order, which also includes *H. brasiliensis* (rubber tree, Euphorbiaceae) and the Passifloraceae family. Rubber tree sap (latex) is particularly rich in T3s, especially δ-T3 and α-T3 [19]. While T3s have not been investigated in Passiflora leaves, the seed oil is known to contain T3s, including δ-T3, but not α-T3 [76,77].

The differentiating trend towards γ-T3 (*Hypericum*) or δ-T3 (*Clusia*) may be a result of different 2-methyl-6-geranylgeranyl-1,4-benzoquinol (MGGBQ, a direct precursor of δ- and β-T3) methyl transferase (MT) activity and VTE3 expression in the two genera. MT adds a methyl group to MPBQ, creating 2,3-dimethyl-6-geranylgeranyl-1,4-benzoquinol (DMGGBQ), the precursor to γ-T3. Upon further methylation, γ-T3 is converted into α-T3. An analogous biosynthetic pathway applies to tocopherols, but phytyl pyrophosphate (PPP) is used instead of geranylgeranyl pyrophosphate (GGPP) [1]. There is an apparent genus-dependent affinity towards MGGBQ or DMGGBQ for the synthesis of T3s, but not for the synthesis of Ts, since α-T dominates over γ-T in all investigated samples. Additionally, in *H. olympicum*, γ-T3 is further methylated into α-T3, while in *C. tocuchensis*, *C. flumensis,* and *H. xylosteifolium* Ts are heavily favored (PPP instead of GGPP).

Tocochromanol contents were highly variable between species in both genera, especially α-T content in *Clusia* spp. leaves, ranging between 10.97 and 112.72 mg 100 g^−1^ (more than ten times), while in *Hypericum* species, other tocochromanol contents were more variable. Since α-T was distinctly dominant in *Clusia* samples, the TT content also ranged by more than tenfold. Tocochromanol proportions were more similar between species in a given genus. In *Hypericum*, the proportions varied more for tocochromanols with higher concentrations, while *Clusia* sample proportions were more consistent. Apart from having generally lower α-T content than *Clusia* leaves, *Hypericum* leaves contained more α-T3 and γ-T3, which were low in *Clusia* leaves. The mean contents of other tocochromanols were similar between samples (Figure 4).

Compared to current industrial sources of tocotrienols, *Hypericum* and *Clusia* leaves are underwhelming. The leading raw materials for tocochromanol extraction are palm fruit and annatto. The proportions of tocotrienol in palm fruit and tocochromanols in oil can be 55–95% and are most often 65–85% [59]; tocochromanol content is 87.9 mg 100 g^−1^, and γ-T3 and α-T3 have the highest proportions [16]. Supercritical CO_2_ annatto seed extract can contain up to 3.13 mg 100 g^−1^ of δ-T3 and 0.39 mg 100 g^−1^ of γ-T3 [78].

However, supplementation of tocopherols and tocotrienols is not very common, and the main dietary sources of tocopherols are vegetable oils, while whole grain products make the largest contribution to tocotrienol intake [17] due to the relatively high tocotrienol content in wholegrain products. Bran oils are some of the richest tocotrienol-containing products. Rye, spelt, wheat, and barley bran oils contain around 346.9, 365.4, 331.7, and 325 mg 100 g^−1^, and wheat germ oil contains around 328 mg 100 g^−1^ tocochromanols, composed primarily of α-T3 and β-T3 [16]. The tocochromanol content in whole grains ranges from 1.64 to 4.16 mg 100 g^−1^ [79]. Other oils that are particularly rich in tocochromanols include sea buckthorn oil, which is particularly rich in α-T, American cranberry seed oil, which is rich in tocotrienols, and common oak acorn oil, with a total tocochromanol content of 452 mg 100 g^−1^ oil [16].

Principal component analysis (PCA) returned principal components PC1 and PC2, which explained 85.66% and 10.01% of the variance, respectively, for a total of 99.92%. PC1 explained an overwhelming majority of the variance and had a strong positive loading with α-T (0.988) and a smaller positive loading with δ-T3 (0.147), while other tocochromanol loadings were close to zero (between −0.1 and 0.1). While minor, PC2 had strong negative loadings with α-T3 (−0.877) and δ-T3 (−0.474). Across PC1 and PC2, α-T overwhelmingly had the highest contribution, followed by α-T3 and δ-T3. Datapoint groups for species in the two genera overlap according to PC1 and PC2 coordinates, with two exceptions, *H. olympicum* and *C. minor*, which are outliers. This is on account of high α-T3 content in *H. olympicum* and δ-T3 in *C. minor*. However, *C. minor* differs relatively little from other representative leaves in terms of tocochromanol content or composition. Species data points are plotted according to their PC2 and PC2 scores in Figure 5. 

## 3. Materials and Methods

### 3.1. Reagents

Ethanol, methanol, ethyl acetate, *n*-hexane (HPLC grade), pyrogallol, sodium chloride, and potassium hydroxide (reagent grade) were purchased from Sigma-Aldrich (Steinheim, Germany). Ethanol (96.2%) for leaf sample saponification was obtained from SIA Kalsnavas Elevators (Jaunkalsnava, Latvia). Standards of tocopherol homologs (α, β, γ, and δ) (>98%, HPLC) were obtained from Extrasynthese (Genay, France), while tocotrienol homologs (α, β, γ, and δ) (>98%, HPLC) were obtained from Cayman Chemical (Ann Arbor, MI, USA).

### 3.2. Plant Material

The cuttings of seven *Hypericum* species: *H. aegyptiacum* (item no. 20040161*B), *H. calycinum* (item no. 10005724*A), *H. lancasteri* (item no. 19862283*A), *H. empetrifolium* (item no. 20100244*A), *H. perforatum* (item no. 20190379*B), *H. xylosteifolium* (item no. 20190402*B), and *H. olympicum* f. *minus* ‘Sulphureum’ (item no. 20110105*D), and five *Clusia* species: *C. fluminensis* (item no. 20130339*A), *C. minor* (item no. 20130341*A), *C. odorata* (item no. 19640298*A), *C. palmicida* (item no. 20130340*B), and *C. tocuchensis* (item no. 20130342*A) were kindly donated by Cambridge University Botanic Garden, 1 Brookside, Cambridge, CB2 1JE, UK. The cuttings were originally intended for plant propagation. The phytosanitary certificate no. UK/GB/E&W/2022/1764253336568 was obtained on 11 May 2022. The cuttings were sent via courier on 12 May 2022 and received on 19 May 2022. The cuttings were packed in a turgid state, sprayed with water to keep the plants alive, and shipped to analytical facilities in sealed bags. All received plants were in good condition, with no rot or mold. For each species, we received at least three cuttings. Plants were prepared for propagation by external cutting and removing most of the leaves. The leaves of each cutting represented individual biological replicates. For each plant species, three biological replicates were obtained. Each biological sample of leaves was separately frozen at −80 °C and stored at this temperature for a week, and subsequently freeze-dried using a FreeZone freeze–dry system (Labconco, Kansas City, MO, USA) at a temperature of −51 ± 1 °C in a vacuum (pressure below 0.01 mbar) for 48 h. The dried plant material for each sample (1 − 10 g) was completely powdered using an MM 400 mixer mill (Retsch, Haan, Germany). The milling parameters were 30 Hz for 60 sec. The obtained 5 μm final fine powder (according to the manufacturer) was used directly for tocopherol and tocotrienol homolog extraction, as described in paragraph 3.3 below. Dry mass was measured gravimetrically. The time of leaf sample processing (from receiving plant material until HPLC analysis) was one month.

### 3.3. Saponification and n-Hexane–Ethyl Acetate Extraction Protocol

The saponification protocol was performed as described earlier [80]. This protocol (saponification) is the most efficient method of sample preparation for tocochromanol determination due to the superior extractability of both tocopherols and tocotrienols from the plant matrix [60]. The recovery of tocopherols and tocotrienols from the leaves of the genera *Clusia* and *Hypericum* was not tested. A 0.1 g of powdered leaf sample was placed in a 15 mL glass tube with a screw cap. Then, 0.05 g of pyrogallol was added to prevent oxidation of tocopherols and tocotrienols. The mixture was sequentially supplemented with 2.5 mL of 96.2% ethanol and mixed. The process of saponification was induced by adding 0.25 mL of 60% (*w*/*v*) aqueous potassium hydroxide. The glass tube was immediately closed with a screw cap, mixed for 10–15 s using vortex REAX top (Heidolph, Schwabach, Germany) with vibration frequency rates of up to 2500 rpm, and subsequently subjected to incubation in a water bath at 80 °C. After 10 min of incubation, the sample was mixed again for 10–15 s at 2500 rpm using the vortex REAX top. After 25 min of incubation to stop/slow down the process of saponification, the sample was cooled immediately in an ice water bath for 10 min. The process of tocopherol and tocotrienol homolog extraction was initiated by adding 2.5 mL of 1% (*w*/*v*) sodium chloride to the glass tube to lower the surface tension between the two non-miscible solvents (hydro ethanol and *n*-hexane–ethyl acetate) and mixing for 5 s at 2500 rpm using the vortex REAX top. Then, 2.5 mL of *n*-hexane–ethyl acetate (9:1; *v*/*v*) was added to extract the tocopherol and tocotrienol homologs and mixed for 15 s at 2500 rpm using the vortex REAX top. After mixing with the organic solvent mixture (*n*-hexane and ethyl acetate), the sample was centrifuged for 5 min (1000× *g*, at 4 °C). The organic layer, containing *n*-hexane and ethyl acetate, was moved to a 100 mL round bottom flask. The extraction residues were re-extracted in a fresh portion of 2.5 mL of *n*-hexane–ethyl acetate (9:1; *v*/*v*), as described above. Re-extraction was performed two times. The organic layer from the initial extraction and the two re-extractions was collected and combined in the same 100 mL round bottom flask and evaporated in a vacuum rotary evaporator Laborota 4000 (Heidolph, Schwabach, Germany) at 40 °C until fully dry. The obtained thin film layer on the bottom of the flask was dissolved in 1 mL of ethanol (HPLC grade) and transferred to a 2 mL analytical glass vial.

### 3.4. Tocopherol and Tocotrienol Determination by RP-HPLC-FLD

Tocopherol and tocotrienol homologs were determined using a previously developed and validated method [81]. The tocochromanol analysis was performed using reverse-phase high-performance liquid chromatography with fluorescent light detector (RP-HPLC-FLD) via a HPLC system (Shimadzu Corporation, Kyoto, Japan) consisting of a pump (LC-10ADvp), a degasser (DGU-14A), a low-pressure gradient unit (FCV-10ALvp), a system controller (SCL-10Avp), an auto-injector (SIL-10AF), a column oven (CTO-10ASvp), and a fluorescence detector (RF-10AXL). The chromatographic separation of tocopherol and tocotrienol homologs was carried out on the Luna PFP(2) (pentafluorophenyl phase) column with the following parameters: particle morphology—fully porous; particle size—3 µm; column length—150 mm; and column ID—4.6 mm secured with a guard column of length 4 mm and ID 3 mm (Phenomenex, Torrance, CA, USA). Chromatographic analysis was performed under the isocratic conditions as follows: mobile phase—methanol with water (93:7; *v*/*v*); flow rate—1.0 mL/min; column oven temperature—40 °C; room temperature—22 ± 1 °C. The total chromatography runtime was 13 min. Identification and quantification were performed using a fluorescence detector at an excitation wavelength of 295 nm and emission wavelength of 330 nm. The quantification was conducted based on the calibration curves obtained from tocopherol and tocotrienol standards.

### 3.5. Statistical Analysis

The results were presented as mean ± standard deviation (*n* = 3) of three independent replications (biological samples) of the plant material. The means ± standard deviations were calculated in Microsoft Excel (Version 1808, Microsoft Office). PCA was performed using the open-source R libraries psych and factoextra in RSutdio software build 764 “Kousa Dogwood” Release (cf37a3e5488c937207f992226d255be71f5e3f41, 2024-12-11) (Posit Software, PBC) without data scaling to avoid noise from minor compounds using species’ mean tocochromanol contents. Visualizations were produced using MS Excel and R open-source libraries ggplot2, ggthemes, and ggrepel.

## 4. Conclusions

Tocochromanols (tocotrienols and tocopherols) were investigated in the leaves of *Hypericum* and *Clusia* plant species. In line with the photosynthetic tissue of most plants, α-T was present at the highest concentration in all studied samples. The leaves contained tocotrienols in notable concentrations, and either δ-T3 (in *Clusia*) or γ-T3 (in *Hypericum*) had higher concentrations. The findings are somewhat corroborated by tocotrienol presence in other better-studied Malpighiales order families such as Euphorbiaceae and Passifloraceae, and indicate the observed low prevalence of tocotrienols in plants and their photosynthetic tissue may be a result of sample selection rather than actual rare occurrence. Given the complex profile of tocochromanols in the leaves of the *Clusia* and *Hypericum* genera, broader species and tocochromanol selection and tocochromanol identity verification with mass spectrometry (MS) and/or nuclear magnetic resonance (NMR) tools are advisable.

Additionally, the concentration of α-T was higher than previously observed in leafy greens, and total tocochromanol and tocotrienol contents were close to those of vegetable oils. Therefore, *Hypericum* and *Clusia* leaves are potential sources of tocochromanols, especially *Hypericum*; however, the plants contain other bioactive substances that are co-extractable with tocochromanols and can adversely interact with medications. To use *Hypericum* as a source of tocotrienols, meticulous purification would be necessary.

## Figures and Tables

**Figure 1 molecules-30-00709-f001:**
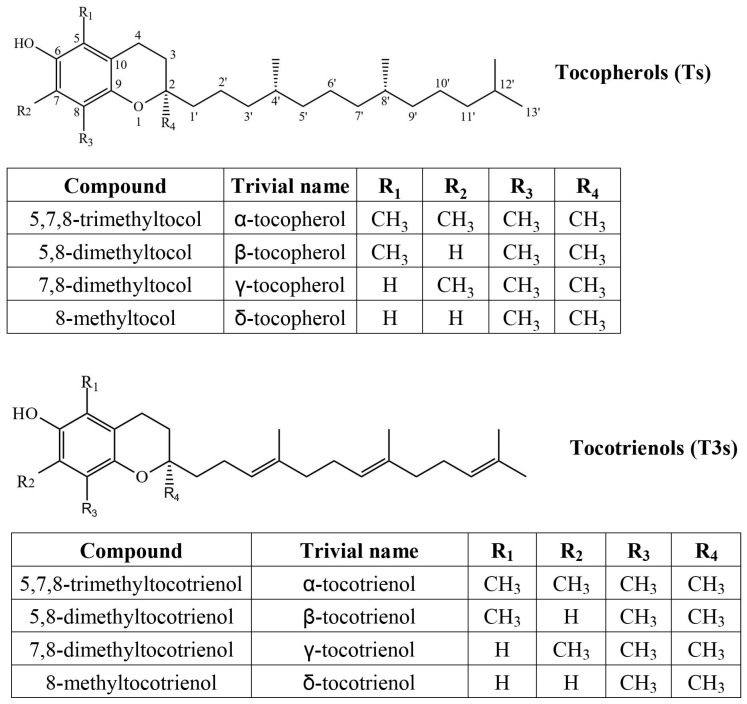
Chemical structures of four tocopherol (T) and four tocotrienol (T3) homologs.

**Figure 2 molecules-30-00709-f002:**
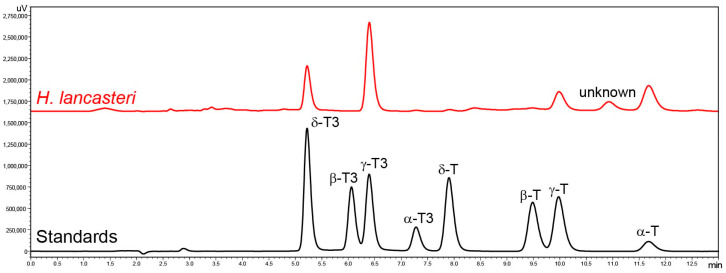
Chromatograms of the separation of tocotrienol (T3) and tocopherol (T) homologs (α, β, γ, and δ) in standards and the leaves of *H. lancasteri* by RP-HPLC-FLD.

**Figure 3 molecules-30-00709-f003:**
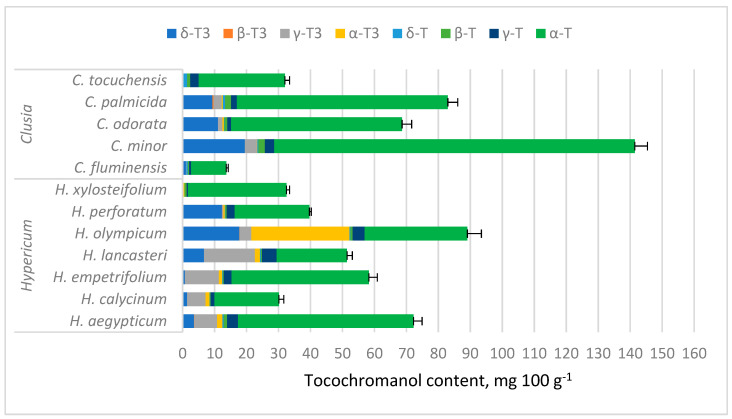
Tocochromanol content in *Hypericum* and *Clusia* species’ leaves. Data are presented as stacked means of three replications of each tocochromanol ± standard deviation of total tocochromanol content.

**Figure 4 molecules-30-00709-f004:**
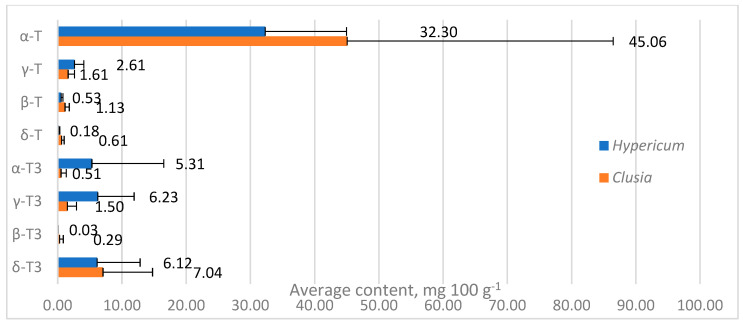
Mean contents of tocochromanols in *Hypericum* and *Clusia* genera. Data are presented as mean tocochromanol content + standard deviation.

**Figure 5 molecules-30-00709-f005:**
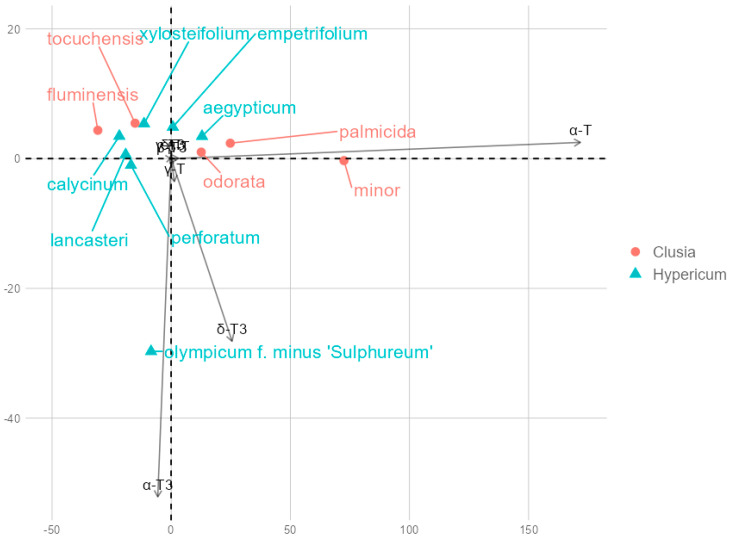
Principal component biplot based on tocochromanol mean contents. Overlapping contents include β-T and T3, δ-T, γ-T3, which had insignificant loadings with PC1 and PC2.

## Data Availability

The data used to support the findings of this study are available in the Appendix A and from the corresponding author upon request.

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
