# Peer review of "Tocochromanols in the Leaves of Plants in the Hypericum and Clusia Genera"

_molecules, 2025, doi:10.3390/molecules30030709_

Round 1
Reviewer 1 Report
Comments and Suggestions for Authors
The article entitled “Tocochromanols in Hypericum and Clusia genus species’ leaves” presents a report on the variations in tocochromanol types (α-, β-, γ-, and δ-tocotrienol and tocopherol) and contents within the leaves of 12 distinct species belonging to the Hypericum (Hypericaceae) and Clusia (Clusiaceae) genera. It was concluded that the Hypericum genus serves as a valuable source of tocotrienols.
Nevertheless, there are several issues need to be clarified or revised before accepting, which are elaborated as follows:
1. The selection and usage of keywords in the article require careful consideration. For instance, keywords like ‘Guttiferaceae’, ‘photosynthetic’ and ‘tocol’ might need to be evaluated further for their appropriateness and relevance.
2. The chemical structures of α-, β-, γ-, and δ-tocotrienol and tocopherol are required to be provided.
3. In light of the description in section 3.3, how does the extraction yield of tocochromanols in each plant leaves fare? A comprehensive summary table (or other suitable formats, as deemed necessary) containing this pertinent information should be furnished in the Supplementary Material to facilitate a more in-depth understanding and reference.
4. Please also provide the original HPLC chromatograms of tocotrienols and tocopherols detection in the leaves of each plant (like Figure 1) as Supplementary Material.
5. The arrows and names in Figure 4 are not clearly depicted, causing difficulties in understanding the figure.
6. There is a lack of uniformity in the reference formats used throughout the article.
Author Response
We sincerely thank you for all the comments, remarks, and suggestions that have contributed to enhancing the manuscript and its scientific quality. The graphical abstract, manuscript, and supplementary materials have been improved accordingly. Provided changes are marked in red font. For literature we used references manager software therefore changes are not highlighted.
Reviewer 1.
The article entitled “Tocochromanols in Hypericum and Clusia genus species’ leaves” presents a report on the variations in tocochromanol types (α-, β-, γ-, and δ-tocotrienol and tocopherol) and contents within the leaves of 12 distinct species belonging to the Hypericum (Hypericaceae) and Clusia (Clusiaceae) genera. It was concluded that the Hypericum genus serves as a valuable source of tocotrienols.
Nevertheless, there are several issues need to be clarified or revised before accepting, which are elaborated as follows:
Comment 1: The selection and usage of keywords in the article require careful consideration. For instance, keywords like ‘Guttiferaceae’, ‘photosynthetic’ and ‘tocol’ might need to be evaluated further for their appropriateness and relevance.
Response 1: Thank you for the comment. “Guttiferaceae” and “tocol” have been removed from the keywords, their inclusion may be considered misleading. “Photosynthetic” is retained for search engine optimization purposes, since leaves are photosynthetic plant organs, but the word “photosynthetic” is not used in the title or abstract.
Comment 2: The chemical structures of α-, β-, γ-, and δ-tocotrienol and tocopherol are required to be provided.
Response 2: Thank you for the comment. A figure of the chemical structures of the tocopherols and tocotrienols was added (page 2).
Comment 3: In light of the description in section 3.3, how does the extraction yield of tocochromanols in each plant leaves fare? A comprehensive summary table (or other suitable formats, as deemed necessary) containing this pertinent information should be furnished in the Supplementary Material to facilitate a more in-depth understanding and reference.
Response 3: Thank you for the comment. Supplementary Material contains a table with the average and standard deviation for each tested sample. The validity of the method has been tested previously, and the relevant paper has been referenced in the manuscript. We did not do additional experiments with the extractability. Additionally, to be clear, in section 3.3 we added following information:
“This protocol (saponification) is the most efficient method of sample preparation for tocochromanols determination due to the superior extractability of both tocopherols and tocotrienols from plant matrix [60]. The recovery of tocopherols and tocotrienols from the leaves of the genera Clusia and Hypericum was not tested” (page 9).
Discussion and comparison of extraction efficiency using different protocols and solvents has been expanded (page 6, top part).
Comment 4: Please also provide the original HPLC chromatograms of tocotrienols and tocopherols detection in the leaves of each plant (like Figure 1) as Supplementary Material.
Response 4: Thank you for the comment. All chromatograms have been provided in the Supplementary Material, including standards of tocopherols and tocotrienols for comparing peak fit to test sample. Slight shifting (left and right) is evident is some chromatograms due to very sharp peaks and system sensitivity to any room environment changes (movement of the staff in the room – small temperature fluctuations in an otherwise temperature-controlled room). The provided chromatograms in the main text and supplementary materials were obtained by different column and chromatographic conditions (testing conditions, Kinetex PFP, 250×4.6 mm, 5 µm) due to technical problems with the PC on which the original chromatograms were obtained two years ago. The chromatogram provided in the main text was obtained two years ago before computer system exchange. The chromatograms provided in supplementary materials were obtained about 18 months ago. They were obtained to confirm our previous results using the DAD detector for FLD confirmation. The additional unidentified compound peaks can be seen on those chromatograms. Co-eluted peaks with tocopherol and tocotrienol homologues were split before quantification. The UV spectra for tocopherols and tocotrienols with UV spectra are included in the Supplementary Material. Identification issues were thoroughly discussed, pointing out the weaknesses of the current study (page 4). Additionally, in the conclusion, the following statement: “Given the complex profile of tocochromanols in the leaves of the Clusia and Hypericum genus, broader species and tocochromanol selection is advisable as well as tocochromanol identity verification with mass spectrometry (MS) and/or nuclear magnetic resonance (NMR) tools.” was added (page 10, bottom).
Comment 5: The arrows and names in Figure 4 are not clearly depicted, causing difficulties in understanding the figure.
Response 5: Thank you for the comment. The figure has been modified so that datapoint labels are larger. Minor tocochromanol vector labels could not be improved (page 8).
Comment 6: There is a lack of uniformity in the reference formats used throughout the article.
Response 6: Thank you for the comment. We use in whole manuscript the reference software. We only noticed that the references list contain the full names of the journals as well as their abbreviation. The references have been standardized in accordance with the journal's requirements.
Reviewer 2 Report
Comments and Suggestions for Authors
The author studied the content of tocopherols in the leaves of 12 plants in Hypericum and Clusia genus, and the reviewers had the following questions:
1. Why did the author not provide RP-HPLC-FLD chromatograms of the tocochromanols profile in the leaves of all plants with the standard products?
2. If it is quantitative analysis, the peak area of δ-T3 and γ-T3 in H. lancasteri seems to be greater than α-T, but the content of α-T is higher than δ-T3 and γ-T3.
3. Comparison of liquid chromatogram alone is not reliable enough, why does the author not do LC-MS?
In addtion, in abstract,“while the presence of tocotrienols in the Hypericum genus has not been reported yet.”However, tocopherols have been reported in Hypericum and are also mentioned in the authors' published paper (Industrial Crops & Products, 2025, 224, 120321.)
Author Response
We sincerely thank you for all the comments, remarks, and suggestions that have contributed to enhancing the manuscript and its scientific quality. The graphical abstract, manuscript, and supplementary materials have been improved accordingly. Provided changes are marked in red font. For literature we used references manager software therefore changes are not highlighted.
Reviewer 2.
The author studied the content of tocopherols in the leaves of 12 plants in Hypericum and Clusia genus, and the reviewers had the following questions:
Comment 1: Why did the author not provide RP-HPLC-FLD chromatograms of the tocochromanols profile in the leaves of all plants with the standard products?
Response 1: Thank you for the comment. All chromatograms have been provided in the Supplementary Material, including standards of tocopherols and tocotrienols for comparing peak fit to test sample. Slight shifting (left and right) is evident is some chromatograms due to very sharp peaks and system sensitivity to any room environment changes (movement of the staff in the room – small temperature fluctuations in an otherwise temperature-controlled room). The provided chromatograms in the main text and supplementary materials were obtained by different column and chromatographic conditions (testing conditions, Kinetex PFP, 250×4.6 mm, 5 µm) due to technical problems with the PC on which the original chromatograms were obtained two years ago. The chromatogram provided in the main text was obtained two years ago before computer system exchange. The chromatograms provided in supplementary materials were obtained about 18 months ago. They were obtained to confirm our previous results using the DAD detector for FLD confirmation. The additional unidentified compound peaks can be seen on those chromatograms. Co-eluted peaks with tocopherol and tocotrienol homologues were split before quantification. The UV spectra for tocopherols and tocotrienols with UV spectra are included in the Supplementary Material. Identification issues were thoroughly discussed, pointing out the weaknesses of the current study (page 4). Additionally, in the conclusion, the following statement: “Given the complex profile of tocochromanols in the leaves of the Clusia and Hypericum genus, broader species and tocochromanol selection is advisable as well as tocochromanol identity verification with mass spectrometry (MS) and/or nuclear magnetic resonance (NMR) tools.” was added (page 10, bottom).
Comment 2: If it is quantitative analysis, the peak area of δ-T3 and γ-T3 in H. lancasteri seems to be greater than α-T, but the content of α-T is higher than δ-T3 and γ-T3.
Response 2: Thank you for the comment. This is caused by different fluorescence intensities of the compounds and the concentration of the standard. Their concentration was calculated based on the peak area and standard concentration. The peak area and calculated concentration are therefore not equally proportional for all tocochromanols. Chromatograms can give an illusion of domination of tocotrienols (the highest peaks), however due to the physicochemical properties of various tocochromanols, as well as elution order in isocratic separation in RP-HPLC, tocopherol and tocotrienol peak areas in chromatograms, especially between the homologues α and δ, are not representative of their absolute content.
Comment 3: Comparison of liquid chromatogram alone is not reliable enough, why does the author not do LC-MS?
Response 3: Thank you for the comment. Compounds were verified using standards and the UV spectra, which was typical for all identified tocochromanols. In addition, the presence of α-T and δ-T3 has been verified in Hypericum perforatum leaves using LC-MS before: https://doi.org/10.1021/jf302425z. Each method have own advances and disadvantages. In the previous study https://doi.org/10.1016/j.foodchem.2024.140789 we found that MS indeed helps to solve the issue with identification of trace amounts of tocochromanols. Using isotopic standards solve this issue, however their cost very high. The application of both detectors (DAD and FLD) for identification in many cases is sufficient. A short discussion has been added on using different detectors and possible co-elution with other compounds. Identification issues were thoroughly discussed, pointing out the weaknesses of the current study (page 4; page 6, top). Additionally, in the conclusion, the following statement: “Given the complex profile of tocochromanols in the leaves of the Clusia and Hypericum genus, broader species and tocochromanol selection is advisable as well as tocochromanol identity verification with mass spectrometry (MS) and/or nuclear magnetic resonance (NMR) tools.” was added (page 10, bottom).
Comment 4: In addition, in abstract, “while the presence of tocotrienols in the Hypericum genus has not been reported yet. “However, tocopherols have been reported in Hypericum and are also mentioned in the authors' published paper (Industrial Crops & Products, 2025, 224, 120321.)
Response 4: Thank you for the comment. The manuscript was prepared before the publication of the 2025 article in question. The abstract has been updated (page 6, top).
Reviewer 3 Report
Comments and Suggestions for Authors
The manuscript entitled "Tocochromanols in Hypericum and Clusia genus species’ leaves" by Mišina et al., studied tocochromanol (α-, β-, γ-, and δ-tocotrienol and tocopherol) contents in the leaves of different species of Clusia and Hypericum using HPLC.
Specific Concerns:
1. A comparison of different plants which are abundant in tocochromanol contents with the current study would be useful.
2. Is there any significant advantages of Clusia and Hypericum compared to the other candidates?
3. What are the rationale for using leaves for the extraction of tocochromanol compared to other plant parts?
4. Why certain tocochromanol contents are mostly abundant in some species compared to others.
5. What are the physiological significance of elevated tocochromanol levels in certain plants compared to others?
6. Does purification of the tocochromanols are among the future objectives of the study?
7. A comparison or table regarding the tocochromanol contents resulting from various solvent extraction methods should be useful.
8. Does the extracts have bio-active properties? Does the authors tested any functional activity of the extracts such as antioxidant properties?
Comments on the Quality of English Language
Manuscript should be checked thoroughly for grammatical and typographical errors.
Author Response
We sincerely thank you for all the comments, remarks, and suggestions that have contributed to enhancing the manuscript and its scientific quality. The graphical abstract, manuscript, and supplementary materials have been improved accordingly. Provided changes are marked in red font. For literature we used references manager software therefore changes are not highlighted.
Reviewer 3.
The manuscript entitled "Tocochromanols in Hypericum and Clusia genus species’ leaves" by Mišina et al., studied tocochromanol (α-, β-, γ-, and δ-tocotrienol and tocopherol) contents in the leaves of different species of Clusia and Hypericum using HPLC.
Comment 1: A comparison of different plants which are abundant in tocochromanol contents with the current study would be useful.
Response 1: Thank you for the comment. The original draft did contain some relevant examples of plants abundant in tocochromanol, however, according to the recommendation, we supplemented it with some more (page 6, bottom, page 7, bottom, page 8, top).
Comment 2: Is there any significant advantages of Clusia and Hypericum compared to the other candidates?
Response 2: Thank you for the comment. The tocotrienol contents were relatively high for plant leaves, though not as high as some specialized oils. The study was primarily aimed at expanding the existing knowledge on tocochromanols in plant materials.
Comment 3: What are the rationale for using leaves for the extraction of tocochromanol compared to other plant parts?
Response 3: Thank you for the comment. As described in the section "3.2. Plant material," the Cambridge University Botanic Garden, Cambridge, UK, kindly provided us with cuttings originally intended for plant propagation. The cutting had leaves and generally untouched stems, with no flowers.
Comment 4: Why certain tocochromanol contents are mostly abundant in some species compared to others.
Response 4: Thank you for the comment. Tocochromanol contents vary significantly between different species in a single family, especially if the plant produces significant amounts of tocotrienols. The high α-T3 content in H. olympicum f. ‘Sulphureum’ is difficult to explain, and may be an anomaly in the variety itself, but we do not possess the tools for metabolomic studies. However, we and others have observed similar anomalies in other plant families e.g. Euphorbia (unpublished data).
Comment 5: What are the physiological significance of elevated tocochromanol levels in certain plants compared to others?
Response 5: Thank you for the comment. The different tocochromanol contents and profiles and first and foremost dictated by the species. While tocochromanol biosynthesis can be affected by environmental factors, the plants were sourced from the same source, and grown and transported under similar conditions. Because of the limited amount of information on tocochromanol contents in Hypericum and Clusia leaves, it is not yet possible to tell if the tocochromanol profiles are typical for the species, or a result of genetic metabolic abnormalities, or environmental factors.
Comment 6: Does purification of the tocochromanols are among the future objectives of the study?
Response 6: Thank you for the comment. Yes, we are working on tocochromanol extraction from Hypericum using green technologies and solvents as a separate study with plant material cultivated in our Institute garden.
Comment 7: A comparison or table regarding the tocochromanol contents resulting from various solvent extraction methods should be useful.
Response 7: Thank you for the comment. While different solvents and extraction methods were not tested within this study, some contextual discussion has been added to the manuscript (page 6, top). Experiments could not be performed as part of this study due to limited plant material.
Comment 8: Does the extracts have bio-active properties? Does the authors tested any functional activity of the extracts such as antioxidant properties?
Response 8: Thank you for the comment. Owing to their tocochromanol contents, the extracts would have antioxidant properties. However, the extracts may also have other components with antioxidant properties. Furthermore, tocochromanols have different antioxidant activities at the same dose, and their antioxidant activity changes differently depending on dose. We did not consider analysing any bioactive properties. Such a study would make more sense if would be performed on-line during detection to the antioxidative contribution of each extract constituent. Unfortunately, we do not have such a system.
Comments on the Quality of English Language
Manuscript should be checked thoroughly for grammatical and typographical errors.
Thank you for the comment. We sincerely apologize for the issue and would greatly appreciate any specific feedback regarding the improvements that are needed. The language was improved by native speaker Dr. Arianne Soliven.
Round 2
Reviewer 2 Report
Comments and Suggestions for Authors
The authors have generally revised and improved the manuscript well. The reviewer suggest that the manuscript can be further considered accepting in present form.